# Development of Perylene-Based Non-Fullerene Acceptors through Bay-Functionalization Strategy

**DOI:** 10.3390/ma13092148

**Published:** 2020-05-06

**Authors:** Keisuke Fujimoto, Masaki Takahashi, Seiichiro Izawa, Masahiro Hiramoto

**Affiliations:** 1Department of Applied Chemistry, Faculty of Engineering, Shizuoka University, 3-5-1 Johoku, Naka-ku, Hamamatsu, Shizuoka 432-8561, Japan; fujimoto.keisuke@shizuoka.ac.jp; 2Institute for Molecular Science, 5-1 Higashiyama, Myodaiji, Okazaki, Aichi 444-8787, Japan; izawa@ims.ac.jp (S.I.); hiramoto@ims.ac.jp (M.H.)

**Keywords:** perylene, non-fullerene acceptor, organic solar cell

## Abstract

Perylene has had a tremendous impact in the history of material research for the molecular semiconductors. Among numerous derivatives of this polyaromatic hydrocarbon, perylene diimide (PDI) represents a promising class of organic materials envisioned as non-fullerene acceptors (NFAs) for the practical organic photovoltaic (OPV) applications due to their enhanced photo- and thermal stability and remarkably high electron affinity, some of which realize band-like transport properties. The present review guides some of the representative achievements in the development of rationally designed PDI systems, highlighting synthetic methodologies based on bay-functionalization strategies for creating well-designed molecular nanostructures and structure-performance relationship of perylene-based small molecular acceptors (SMAs) for the photovoltaic outcomes.

## 1. Introduction

### 1.1. Discovery of Perylene Dyes

Perylene, a primitive type of polyaromatic hydrocarbon comprising five benzene rings, has had a tremendous impact on the history of material research for the molecular semiconductors, since the discovery of electrical conductivity of perylene-bromine charge-transfer (CT) complex by Akamatsu, Inokuchi, and Matsunaga in 1954 [1]. Because of the electron-rich nature, the native perylene acts as an electron donor, playing a p-type semiconductor material in contact with electron-deficient molecules such as bromine. At this consideration, one may expand the potential molecular diversity around the perylene scaffold to find an electron-accepting counterpart by introducing electron-withdrawing elements into the aromatic nucleus [2].

Representative examples of such molecules include 3,4,9,10-perylenetetracarboxylic acid (PTCA) and its dianhydride (PTCDA), which contain carboxylic functionalities at the so-called peri positions (Figure 1). According to the literature, PTCA has appeared as a vat dye in a US patent published in 1924 [3], while PTCDA has been available later in 1933 [4]. These compounds have been manufactured by oxidative coupling of naphthalene 1,8-dicarboxylic acid or its derivatives such as a 1,8-monoimide variant, which the contemporary production by commercial suppliers still relies on [5]. Here, it is of interest to note that a monoimide derivative of PTCA has been incidentally obtained as a consequence of incomplete conversion to the desired product through the synthetic process [6]. Afterward, diimide derivatives of PTCA, which we simply call perylene diimides (PDIs), have been intentionally produced by condensation of PTCA with amines to dye cotton and wool fabrics with Bordeaux-red shades as described in a US patent published in 1923 [7], ten years after the first discovery in 1913 by Kardos [8,9]. This type of reaction has been applied to the synthesis of different electron-deficient perylene systems, namely perylenetetracarboxylic dibenzimidazoles (PTCBIs), wherein 1,2-diamines undergo imidation and the following imination into benzimidazoles that adopt extended π-conjugation in the planar perylene systems [10]. Because of high thermal and chemical stability, these perylene derivatives were employed as a robust coloring matter in the early stages.

### 1.2. Application for Organic Photovoltaics

Later on, researchers and engineers have widely applied these materials in the areas of organic molecular electronics. Among these applications, particularly worthy of note is the use of the perylene dyes as important constituents in organic photovoltaics (OPVs) [11]. The OPV devices, which potentially employ organic semiconductor materials for light-harvesting to facilitate charge separations, have been developed since the late 1950s [12]. In earlier studies, researchers have made a primitive type of OPVs based on a single-layer structure, wherein phthalocyanines gave a small but significant level of photovoltaic performance [13]. Despite the prominent success, the poor efficiency of photocurrent generation has been a major problem for the single-layer devices. One of the biggest breakthroughs to resolve this issue has been achieved by the invention of organic–organic heterojunction systems innovated by Tang (Figure 2) [14]. In his work, he set up a two-layer organic cell made through sequential vacuum evaporation of copper phthalocyanine (CuPc) and PTCBI onto an indium tin oxide (ITO) substrate. The work has demonstrated that the PTCBI layer obviously served as an electron-acceptor, extracting electrons from CuPc at their interfaces, to enhance the photovoltaic performance, demonstrating that the nature of organic–organic interfaces is the principal determinant of photovoltaic properties.

The innovative contribution to the next evolution of the OPV technologies has been made by Hiramoto and coworkers, who first established the design concept of bulk heterojunction (BHJ) blends. Their OPV devices have been fabricated with sublimated metal-free phthalocyanine (H_2_Pc) donor and *N,N′*-dimethyl-substituted PDI (Pigment Red 179) acceptor to deposit three organic layers, which contain an interlayer (i-layer) of co-deposited donor/acceptor pigments between homogeneous layers of the two respective constituents. This work has ascertained that the pigment mixing enlarges the donor/acceptor interface to dramatically improve the photogeneration efficiency (Figure 2) [15]. The development of the three-layered p-i-n structure, where p and n denote the donor and acceptor layers, respectively, has emphasized the importance of the BHJ active layer to overcome the limitations of OPV performance for the conventional planar heterojunction (PHJ) systems.

### 1.3. Development of Organic Photovoltaic Technology

In the next stage, significant progress has been made in the fabrication protocol by Yu, Heeger, and coworkers. They adopted a solution process with the aim of developing practical and cost-effective technologies. This alternative way allows one to use non-sublimable organic semiconductors such as conjugated polymers and thermally less robust molecules [16]. Furthermore, the work has highlighted the potential utility of buckminsterfullerene (C_60_) derivatives as the electron acceptor (i.e., n-type semiconductor) because of their high electron conductivity. In the actual systems, two types of soluble phenyl-C61-butyric acid methyl esters, namely [6,6]- and [5,6]-PC_61_BMs, were employed to fabricate the BHJ blend films with a donor of poly(2-methoxy-5-(2′-ethylhexyloxy)-1,4-phenylene vinylene), denoted as MEH-PPV, which were deposited by spin-cast from xylene and 1,2-dichlorobenzene solutions (Figure 3). Accordingly, the solution-processed devices provided comparably high efficiencies with regard to carrier collection and energy conversion, as has also been demonstrated in self-organized discotic liquid crystals composed of hexaphenyl-substituted hexabenzocoronene (HBC-PhC_12_) and *N*,*N′*-bis(1-ethylpropyl)-substituted PDI by Friend, Müllen, and coworkers [17]. Because of the superior versatility of organic materials and good processability combined with low-cost operation, the solution processing method has been accepted as the standard technique for contemporary OPV devices [18].

Thus, the OPV technology has recently emerged as credible and prominent future power sources, because of the economic and environmental advantages to offer low production disposal costs for processing hazardous-element-free materials, unlike the inorganic and perovskite counterparts [19,20]. Furthermore, the technological applications of the solution processing methods provide attractive options to fabricate semitransparent devices that allow for harnessing solar power through colored windows and roofs of buildings and vehicles [21]. The commercial scale levelized cost of electricity (LCOE) represents a measure of power sources to compare different methods of electricity generation. According to this estimate, mass-manufactured OPV modules with the power conversion efficiency (PCE) of 5% and the operating life time of 5 years are predicted to produce an LCOE under $0.13/kWh, which is comparable to that of the current coal-based electricity generating systems [22]. It deserves to be mentioned that the OPVs, being compatible with large-scale roll-to-roll print manufacturing when fabricated on plastic substrates, combine additional advantages of flexibility and light weight, making them suitable as a next-generation avenue for generating portable and renewable energies [23,24].

### 1.4. Non-Fullerene Acceptors for Organic Solar Cells

As described above, the selection of organic semiconductors is crucial for the development of high-performance organic solar cells (OSCs). In contrast to a wide variety of the p-type semiconductor donor materials available nowadays, there is a very limited option of the n-type counterparts [25,26]. As a matter of fact, the current OSCs are being made with π-conjugated donor polymers that exhibit efficient hole transporting properties and good chemical compatibility with small molecular acceptors (SMAs) to tailor microcrystalline film morphologies of the polymer/SMA BHJ blends [27]. The earlier efforts have focused on the use of the fullerene-based materials as a potential SMA, mainly because of their excellent electron transport properties with high chemical stability, accompanied by favorable polymer compatibilities [28]. Despite the highly productive results obtained in many research works, the design-to-device approach based on fullerene acceptors suffers from inherent poor absorbance in the UV-visible region and limited tunability in terms of synthetic flexibility, cost efficiency, and amenability to scale-up manufacturing [29].

Thus, non-fullerene acceptors (NFAs) have recently emerged as promising alternatives to the conventional fullerene-based acceptors [30]. Researchers have synthesized a vast number of rationally designed NFAs with superior light-harvesting ability in the visible and near-infrared (NIR) regions, which have realized dramatically improved PCEs [31]. Indeed, the highest PCE of the state-of-the-art OSC based on Y6 has already reached over 17%, when coupled with complementary polymeric donor PM6 (Figure 4) [32,33]. Further research efforts with a Y6 derivative have made the most successful achievement in this field with remarkably enhanced PCE approaching 18% [34]. However, these types of NFA materials still face performance degradation because of their susceptibility to moisture and oxygen, offering less long-term durability [35,36] and suffer from synthetic complexity [32]. Therefore, it is highly desired to find chemically robust alternatives that perform well under a range of circumstances. At this point, the perylene-based NFAs have been envisioned as the other candidate for the practical OPV applications because of their enhanced photostability, thermal stability, and remarkably high electron affinity, some of which realize band-like transport properties [37,38,39].

### 1.5. Chemical Modification of Perylene Systems

Chemical modification of the perylene backbones, such as PTCDA and PDI, produces rich structural diversity to offer a broad range of functional properties [40]. The chemical modification is classified into functionalization at so-called bay (1,6,7,12-) positions and ortho (2,5,8,11-) positions (Figure 1). Although recent efforts in the ortho-functionalization endowed the perylene-based materials with a wide structural and functional diversity, such processes may be ineffective to tune the electronic properties of the perylene π-systems, because the ortho carbon atoms have inherent poor reactivity and relatively small contribution of the frontier molecular orbitals (FMOs) which determine the photophysical properties of the perylene nucleus [41,42].

Instead, bay-functionalization is more suitable for fine-tuning of optical and electronic properties of the perylene π-system, because the bay-carbons possess more electronic density of the highest occupied molecular orbital (HOMO) to facilitate electrophilic substitution reactions and alteration of the electronic properties of the perylene π-system [43,44]. Conventional methods for the bay-functionalization rely on halogenation of the perylene nucleus [45]. The halogenation makes the perylene π-systems reactive toward nucleophiles, enabling introduction of functional groups at the bay positions [46]. Nevertheless, only limited methods are available for the halogenation reactions, most of which have low controllability with regard to precise location and number of incorporated halogen atoms [47]. As a matter of fact, conventional bay-halogenations require excess chlorine or bromine to yield highly substituted products such as 1,6,7,12-tetrachloride derivatives or inseparable mixtures of 1,7- and 1,6-dibromide ones, respectively (Figure 5) [48,49]. Such a different reactivity can be ascribed to the size of the two halogen atoms; the larger bromine atom prevents further bay-bromination at the opposite shores to afford the dibrominates as regioisomeric mixtures [50].

### 1.6. Synthetic Issues on Perylene Dyes

With this in mind, it is expected that controlling the course of bromination at one of the four bay positions becomes a key process to access monosubstituted perylenes available for various applications and further chemical modification. However, inherent insolubility of the planar perylene molecules has hampered synthesis of the monobromides with practically acceptable yields, thereby causing major limitations in rational design of perylene-based materials [51]. To address this issue, many researchers have utilized the “swallow-tail”-like branched alkyl chains on the nitrogen atoms of the imide groups, which bring significant steric hindrance to effectively weaken the excessive degree of π-stacking and crystallinity [52]. Indeed, the use of the “swallow-tail” substituents solves the solubility problem, enabling efficient monobromination with yields of as high as 57% upon treatment with bromine in dichloromethane (DCM) [53]. With the aid of this strategy, a diverse range of new molecular systems containing the swallow-tailed PDIs have been designed as efficient NFA materials for the OPV applications [54].

In contrast to the synthetic benefits of the solubilizing effects, flexible and bulky nature of the branched alkyl chains causes inevitable degradation of crystallinity or intermolecular interaction, which may cause detrimental effects in OSC performances [55]. On this basis, it is reasonable to find the molecular structures possessing both high solubility and crystallinity for creating ideal perylene-based NFAs. Thus, more flexible and efficient approach to supply the precisely activated perylene materials, independent of structural properties of the side chains, will be the key to achieving the diversity-oriented synthesis of novel highly efficient NFAs. In this regard, operating with more soluble perylene tetracarboxylic esters (PTEs) followed by conducting saponification and imidation to PDIs may represent an alternative strategy to address the above concerns [56]. The objective of this review is to highlight the significant progresses on the development of a diverse range of designed perylene systems as NFA materials. With the intention to clarify what progress has been achieved by structural arrangement of the perylene units in the OPV technology, the primary subject of this review focuses on the perylene-based SMAs developed through bay-functionalization strategies [57].

## 2. Monomeric Series of Perylene-Based Materials for Organic Photovoltaics

The monomeric perylene derivatives have been utilized as efficient n-type acceptor materials in the heterojunction OSCs of many pioneering works, which have been fabricated via vacuum sublimation [14,15]. Because of the fact that the sublimation process cannot be applied to multicomponent molecular systems containing thermally labile covalent bonds, most of the perylene-based NFAs with greater structural complexity are only well suited for the solution processing approach. Relatively small molecular dimensions of the monomeric perylene π-systems may present some disadvantageous situations to result in less ordered face-on stacking of the perylene units upon immersion with p-type donor materials, leading to the formation of inefficient carrier transport pathways in the solution-processed devices [58]. Regarding this issue, it has been well documented that device performance can be improved by the use of high-boiling point co-solvent additives such as 1,8-diiodooctane (DIO) and 1-chloronaphthalene (CN) [59,60]. Although the exact mechanism still remains unclear, these solvent additives have pronounced effects on morphological control accompanied by crystallinity enhancement of active layers to promote photovoltaic performances [61]. In the examples below, one can see that the monomeric architecture of perylene molecules represents one of the simplest and most reliable SMA motifs which ensure significant potential to enhance photoelectric conversion processes.

### 2.1. Bay-Monofunctionalized Perylene Diimides

A primitive molecular engineering of the monomeric PDI system has been investigated by Ma, Wang, and coworkers to probe impacts of different aryl moieties introduced at one bay position on the device efficiency (Scheme 1) [62]. The synthesis of the arylated PDIs was simply achieved as follows. The bromination of *N*,*N*′-bis(2-ethylhexyl)-substituted PDI **1** was carried out under empirical experimental conditions using excessive amount of bromine (more than 60 equiv compared to the substrate material) in chlorobenzene (CB) at 60 °C for 2 days to produce the monobromide **2** in 46% yield. The obtained bromide **2** was subjected to palladium-catalyzed cross-coupling reactions with *p*-propylphenyl, *p*-hexylphenyl, and *p*-nonylphenyl boronic acids to give the three types of arylated PDIs **2.1A**, **3a**, and **3b**, respectively, a small part of which unexpectedly underwent oxidative ring closure to afford the corresponding core-extended PDI variants [63].

Because of the decreased planarity of perylene core caused by steric bulkiness of aryl moieties and/or the decreased molecular symmetry, all the arylated PDIs exhibited much higher solubility than the unsubstituted parent PDI. Among these entries, the *p*-propylphenyl-substituted **2.1A** achieved the highest PCE of 0.77% and remarkably enhanced fill factor (FF) of 0.66, albeit with moderately low open circuit-voltage (*V*_oc_) of 0.63 V and short-circuit current (*J*_SC_) of 1.93 mA cm^−2^ in the OPV device fabricated with a polymeric donor P3HT by using 1% DIO, while the corresponding devices with **3a** and **3b** provided moderate PCEs of 0.37% and 0.39%, respectively. The emission intensities of them were in order of **3a** > **3b** > **2.1A**. The lowest value in **2.1A** suggested the strongest intermolecular interactions which may facilitate efficient electron transportation. Despite weaker intermolecular interaction, **3b** exhibits weaker fluorescence than **3a**, indicating that the long nonyl side chain increases the possibilities of non-radiative deactivation that are envisioned to have detrimental effects on the device efficiency. Furthermore, atomic force microscopy (AFM) and transmission electron microscopy (TEM) revealed that the **2.1A**/P3HT blend film showed well-defined interpenetrating networks with smooth surface and suitable crystalline domains which are beneficial to achieving a high performance in photoelectric conversion. In addition to the following researches of the authors [64], the work offers valuable guidelines for the molecular design of new SMAs; (1) deformation of planarity in molecular backbones to improve the solubility and (2) avoidance of flexible long alkyl chains to form a microcrystalline morphology suitable for photovoltaic performances in the blend film.

### 2.2. Bay-Difunctionalized Perylene Diimides

The most common strategies to obtain bay-difunctionalized PDIs are nucleophilic aromatic substitutions and palladium-catalyzed coupling reactions of perylene dibromides. A number of researchers has adopted a conventional method developed by BASF SE for the preparation of PTCDA dibromide as a synthetic precursor to the dibrominated PDIs [49]. As mentioned above, the dibromination of PTCDA is practically achieved by a treatment of PTCDA with bromine in fuming sulfuric acid at high temperature to produce inseparable regioisomeric mixture of the 1,7- and 1,6-dibromides in high yield. Thus, chemical transformations of the as-prepared PTCDA dibromide deliver the relevant inseparable regioisomeric mixture. Therefore, the bay-difunctionalized PDI materials often encounter troublesome purification due to the structural inhomogeneity. Conventionally, 1,6-difunctionalized PDIs derived from 1,6-dibromides have been thought to be unwanted side products that need to be removed for use in photovoltaic systems. However, it is interesting at this point that recent work has demonstrated the superior photovoltaic properties of 1,6-difunctionalized PDI to its 1,7-isomer, contrary to the expectations that there is a negligible influence of the difference in substitution patterns of the PDI systems on the material properties [65].

As a representative example of bay-difunctionalized PDIs, Sharma, Mikroyannidis, and coworkers have represented potential utility of bay-difunctionalized PDI **2.2A** with aryloxy residues (Scheme 2) [66]. The compound **2.2A** was prepared from PTCDA (**4**) according to a procedure reported by Icli and coworkers [67]. Etherification of PTCDA dibromide **5** with *p*-*tert*-butylphenoxide in refluxing *N,N*-dimethyl formamide (DMF) and the following condensation of the resulting dianhydride **6** with 9-aminoanthracene provided **2.2A** [68]. Unfortunately, the reports lack any details about the regioisomers present in the crude product. The OPV device fabricated in this work employed benzothiazole-cored vinylene derivative BTD-TNP, which possesses a narrow band gap to absorb photons over the whole visible and near-infrared (NIR) range of the solar spectrum. Fortunately, **2.2A** exhibited intense absorption in the green spectral region, where the donor material exhibited relatively weak absorption. As a result, a considerably enhanced PCE of 2.85% was achieved in the BHJ system with a high *V*_oc_ of 0.92 V, a moderate *J*_SC_ of 6.6 mA cm^−2^, and FF of 0.47. Here, the observed *V*_oc_ exceeds that of the corresponding fullerene-based device with PC_70_BM. The authors described that the remarkably large photovoltage should be attributed to the higher lying lowest unoccupied molecular orbital (LUMO) level of **2.2A**, which would be reasonably understood in terms of electron-donating property of the aryloxy units at the bay positions. Thus, the work has amply demonstrated the effectiveness of bay-difunctionalization with electron-donating groups on the significant enhancement of *V*_oc_ parameter of OPV devices.

### 2.3. Fully Bay-Functionalized Perylene Diimides

Chemical functionalization of all the four bay positions offers a potential benefit to shield the strong π–π stacking interactions between perylene cores. This effectively suppresses the excessive aggregation behavior to improve the solubility of the designed compounds as has also been observed for liquid crystalline/mesogenic perylene dyes [69,70,71]. Additionally, significant structural distortion due to steric conflicts between substituents will twist the π-conjugated perylene ring system away from planarity [72]. The out-of-plane deformation induced by the bay-substituents will not only promote good solubility but also result in enhanced intersystem crossing (ISC) to generate long-lived triplet excitons [73]. Recent study indicated that the long-lived triplet state showed longer exciton diffusion length leading to efficient PCEs [74]. The general synthesis of the fully bay-functionalized PDIs has employed the corresponding tetrahalogenated perylene precursors. In practice, the chlorinated perylene molecules have low reactivity toward consecutive palladium-catalyzed cross-coupling reactions, but have enough reactivity toward etherification reactions [75]. On the other hand, the brominated perylenes are more favorable for palladium-catalyzed reactions to take place, despite the limited synthetic availability of the any requisite tetrabromides. In this regard, there remains room for preparing PTCDA tetrabromide upon the bromine treatment of PTCDA. Tian and coworkers have reported that subjection of PTCDA (**4**) to harsher conditions with prolonged reaction time over 2 days at higher temperature to 100 °C led to the formation of the desired tetrabromide **8** in 21% yield as a minor component of the inseparable reaction mixture [76]. Afterward, Zhu, Liu, and coworkers reported high-yielding synthesis of **8** by extending the reaction time to 5 days which featured a satisfactorily high yield of 96% (Scheme 3) [77].

This key intermediate **8** has been applied by Sun and coworkers to prepare a fully bay-functionalized tetraphenyl PDI **2.3A** as a potential NFA [78]. The authors have followed almost the same processes illustrated in the previous work [77], undertaking condensation of dianhydride **7** with cyclohexylamine followed by Suzuki coupling reaction of the obtained diimide **8** with excess phenylboronic acid in the presence of cesium fluoride and silver oxide to obtain a 32% yield of **2.3A** with cyclohexyl moieties on the nitrogen atoms. The density functional theory (DFT) calculation indicated that the perylene core of **2.3A** should be twisted with a dihedral angle of about 15° because of the steric repulsion between the adjacent phenyl appendages. The tetraphenyl PDI **2.3A** exhibited two characteristic absorption peaks at *λ*_max_ ~450 and ~600 nm due to electronic perturbation of the phenyl groups, providing good coverage in the visible spectral range. In addition, the fluorescence quantum yield of **2.3A** was determined as low as 10%, indicating efficient ISC to generate possibly high population of triplet excitons, although there are no commentaries on these subjects.

The PDI **2.3A** has been used to fabricate a solution-processed OSC with a low band gap polymeric donor PTB7-Th that has achieved prominent PCE records over 10% in many other devices [79]. It is of interest to point out that the spectral coverage of the PDI material was shown to complement well with that of PTB7-Th, which is desirable for efficient absorption of the entire visible region of the solar spectrum. Consistent with this expectation, the OPV device produced significantly enhanced PCEs with favorable photovoltaic parameters. Furthermore, the BHJ morphology of the active layer could be engineered by using the processing solvent additive CN to improve the roughness of the film surface. Upon using 1% CN, the BHJ blends showed a smooth morphology, which was reflected by the improved PCE reaching 4.1%, the highest recorded value for the monomeric class of the perylene-based NFAs, together with the optimal device parameters; *V*_oc_ of 0.87 V, *J*_SC_ of 10.1 mA cm^−2^, and FF of 0.41. The work has successfully demonstrated that the fully bay-substituted PDI with twisted geometry of the π-conjugated backbone exhibited weak aggregation tendency in the solid state and broad spectral coverage complimentary with that of PTB7-Th, highlighting the effectiveness of chemical modifications of the bay regions to achieve superior morphological uniformity of BHJ structures and encouraging device performances in OPVs.

### 2.4. Multiply Bay-Functionalized Perylene Diimides

The difficulty in controlling location and number of bay-halogenation has hampered further development of precisely defined PDI derivatives with diverse range of functional groups. As a primitive molecular engineering of the multiple bay-functionalization, Pang, Zhang, and coworkers have performed stepwise nucleophilic aromatic substitutions (S_N_Ars) with a variety of alcohols to produce unsymmetrically 1,7-disubstituted PDIs (Scheme 4) [80]. Treatment of the 1,7-dibrominated PDI **9** and **11** with 5.0 equivalents of the alcohols and potassium carbonate in heating DMF provided monoalkoxy- or monoaryloxy-substituted PDIs **10** and **12**, respectively, in 39–93% yields leaving one bromine atom intact. Further etherification of **12** with a different type of alcohol led to the successful synthesis of an unsymmetrically disubstituted PDI **13**. This success would be attributed to the electronic state differences between the PDI monobromide intermediate and PDI dibromide. Namely, the monobromides **10** and **12** are less reactive than the dibromides **9** and **11**, respectively, because of the significant electron-donating effect of the initially installed alkoxy and aryloxy groups. However, this kind of strategy is only effective for particular substrates and reagents in the situation described above.

Other representative reports to modify the perylene cores with different functional units have focused on preparing unsymmetrical dibromide to discriminate between chemical reactivities of the 1- and 7-positions. Dubey, Jager, and coworkers have successfully demonstrated that 1,7-dibrominated perylene monoimide (PMI) **14** bearing two *n*-butyl ester groups at the opposite peri positions underwent highly regioselective S_N_Ar reactions with equimolar amounts of typical nucleophiles such as 4-*tert*-butylphenoxide and pyrrolidine to give the respective PMI monobromides **15** and **16**, respectively, which were functionalized predominantly at the 7-position (Scheme 5) [81]. Further experimentations on the PMIs obtained in the above reactions revealed an importance of the substituent effect of the initially installed functional groups. Actually, the PMI monobromide **15** with moderately electron-donating aryloxy-group underwent the second S_N_Ar process to give diaryloxy PMI **17**, while the strongly electron-donating pyrrolidine-substituted PDI **16** failed to react with phenoxides.

The authors have further developed the above synthetic strategy to install multiple functional groups at all the bay positions of the PMI ring system [82]. The synthesis utilized 1,6,7,12-tetrachlorinated PMI **18** as a starting compound, which was prepared via several steps from 1,6,7,12-tetrachlorinated PTCDA (Scheme 6). The elaboration of the stepwise bay-functionalization was well manipulated through careful controls of stoichiometric quantities of reagents and reaction temperatures. Indeed, the stronger electron withdrawing character of the imide groups than the ester ones enhances the electrophilic reactivity at the 7- and 12-positions rather than 1- and 6-positions. The potentially reactive electronic structure of **18**, which allowed the reaction operating under milder conditions, made it easier to control the number of aryloxy-substitution. Indeed, single substitution was achieved at the reaction temperature of 65 °C to give the 7-monosubstituted PMI **19** and heating to 95 °C facilitated the formation of the 7,12-disubstituted PMI **20** in high yields.

In the subsequent step, 7,12-diaryloxy PMI **21** as obtained via aryloxylation of **19** with 4-*tert*-butylphenol showed chemical inertness toward the S_N_Ar etherification under the comparable conditions, which addressed a need for converting the PMI backbone to the more electrophilic PDI through imidation. Thus, the PMI **21** was converted to the corresponding PDI **23** via dianhydride **22**. The 1,6-dichlorinated PDI **23** underwent simultaneous S_N_Ar etherifications at the 1- and 6-positions under more vigorous reaction conditions to give the fully aryloxy-substituted PDI **24**. A stepwise installation of different functional groups failed at that stage of the investigation. The work has successfully demonstrated the feasibility of regioselective synthesis of multiply bay-functionalized perylene-based materials, but may also exhibited the inherent limitations in terms of synthetic flexibility, which are imposed by the complexity of the entire synthesis.

At this point, electronic effect of the bay-substituents significantly influences the chemical reactivity of the perylene π-systems. As seen in the above synthetic examples, the presence of electron-donating substituents increases the electronic density on the other bay-carbons, thereby resulting in negative effects on the S_N_Ar reactions to occur. Conversely, it can be assumed that introduction of electron-donating substituents at the bay positions will enhance reactivity of the perylene π-systems toward electrophilic aromatic substitutions (S_E_Ars) like bay-halogenations. Zhan, Yao, and coworkers have successfully demonstrated that a variety of PDIs bearing alkoxy groups underwent efficient bay-brominations under precise direction imposed by the electronic effects of substituents (Scheme 7) [83]. Careful control of the stoichiometric ratio of bromine enabled stepwise brominations of **25** to obtain either 7-monobrominated PDI **26** or 7,12-dibrominated **27** in regioisomerically pure form. Apparently, the above observations can be rationalized by considering the electronic states of the perylene π-systems, where the monobrominated **26** should have less electronic density than the nonbrominated **25** because of the electron-withdrawing ability of the bromine atom, leading to a suppressed subsequent bromination of **26**. The work has opened the way to addressing how to activate the perylene π-systems at specific bay carbon centers through brominations.

Based on the above findings, Ng, Nuckolls, and coworkers have developed the designed structures of regiochemically well-defined PDIs [84]. The synthetic work has shown that the starting methoxy-substituted PDIs **27** and **31** underwent efficient monobrominations in a highly regiospecific fashion to give the relevant monobromides **28** and **32**, respectively (Scheme 8). The researchers applied the synthetic method to access dimeric derivatives of PDI-nanoribbon **30** and PDI-helicene **34**. They performed palladium-catalyzed cross-couplings of **28** and **32** with distannyl ethene and naphthalene diboronic acid pinacol ester to produce dimeric precursors **29** and **33**, respectively. Then, these compounds underwent oxidative photocyclization in an LED flow reactor with iodine to access the fused aromatic systems **30** and **34** [85]. The work has provided significant advancements in the structure-directed synthesis of deliberately designed PDI derivatives, while the described methods still have a problem of the use of inconvenient reagent, namely highly volatile, corrosive, and toxic bromine.

Achieving high controllability of the bay-halogenation, especially bay-bromination, is a key requirement for a continuous and adequate supply of synthetic building blocks for custom-designed perylenes. Practically, addition of a precisely equimolar amount of bromine is very difficult in laboratory scale experiments because of the volatility and high density. In this context, Takahashi, Yoda, and coworkers have utilized *N*-bromosuccinimide (NBS) as a bromine source and highly soluble tetrabenzyl perylene-3,4,9,10-tetracarboxylate (**36**) as a perylene substrate, which is easily prepared by hydrolysis of PTCDA (**4**) and esterification of the resulted PTCA (3**5**) [86]. Application of the catalytic halogenation with ferric chloride (FeCl_3_) in heating acetonitrile (AN), reported by Tanemura and coworkers [87], for bromination of **36** enabled controllable monobromination to provide the monobromide **37** in a high isolated yield of 83%. Here, use of NBS allowed an easy addition of the precise amount of bromine source, which is necessary for the reliable monobromination (Scheme 9) [88]. This new methodology has become an effective tool for synthesizing diversely bay-functionalized perylene molecules with more complex substitution patterns.

Continuous efforts to improve the bay-monobromination of PTE disclosed that the use of DCM, instead of AN as the solvent, facilitated the bromination of **36** even at room temperature to give **37** in an improved yield of 93%. With this encouraging result, Fujimoto, Izawa, Hiramoto, Takahashi, and coworkers have undertaken a progressive bromination/functionalization strategy to precisely tailor structural and electronic properties of the perylene π-systems, which has been directed toward the synthesis of a new class of multiply bay-functionalized PDIs for the application in OPV devices (Scheme 10) [89]. The NBS bromination of aryloxy-substituted PTE **38**, being prepared from the monobromide **37**, underwent regioselectively and stoichiometrically controlled brominations without degradation of the aryloxy appendage; the equimolar use of NBS led to a selective production of 7-brominated PTE, while applying analogous conditions with a total 2.5 equivalents of NBS resulted in the corresponding 7,12-dibromide **39** as an exclusive product. Of particular note is that the following etherification on the 7,12-dibromide took place only at the 7-position to give 1,7-diaryloxy-substituted PTE **40** with a remaining bromine atom at the 12-position. The chemical inertness of the 12-position can be ascribed to steric hindrance imposed by the preinstalled aryloxy group and deactivation of the perylene π-system resulting from increased electronic density due to the aryloxy group. The NBS bromination was still amenable to further bromination on the unsubstituted 6-position of **40** to furnish the corresponding 6,12-dibromide **41**. This fully bay-substituted building block underwent Suzuki-Miyaura cross-coupling with 4-methoxyphenylboronic acid to afford the further arylated PTE **42**. In the final step, all the four benzyl ester groups of **42** were converted to the two *N*-phenyl imide groups upon treatment with excess aniline in melted imidazole, providing the corresponding PDI **2.4A** with C_2h_-symmetrical structure.

The X-ray crystallography of **2.4A** revealed a remarkably twisted structure of the perylene nucleus by an average dihedral angle of 35° due to repulsive interactions between the bay-substituents. Interestingly, this compound showed spectroscopic characteristics similar to the tetraphenyl PDI **2.3A**, exhibiting two absorption peaks at *λ*_max_ ~480 and ~610 nm due to electronic perturbation of the aryl groups. Furthermore, inefficient fluorescence emission with a very poor quantum yield of 2.2% indicated the formation of triplet excitons. This new PDI **2.4A**, which exhibited high solubility in common organic solvents, was then subjected to blending with a chloroform (CF) solution of the polymeric donor PTB7-Th to fabricate a spin-coated BHJ film for the OPV application. In agreement with expectations from smooth and uniform morphology observed at the BHJ surface by AFM, this OPV device achieved enhanced performance with PCE of 1.74%, *J*_SC_ of 4.94 mA cm^−2^, FF of 0.352 under 1 sun illumination, which are considerably higher than those of referential unsubstituted PDI. Furthermore, it should be pointed out that *V*_OC_ obtained from the device with **2.4A** reached 1.00 V, remarkably higher than that of either the referential PDI (0.75 V) or the structurally similar **2.3A** (0.87 V). Taking into account the optical band gap of 1.57 eV obtained from the BHJ blend of **2.4A**/PTB7-Th, voltage loss (*V*_loss_) of the device was estimated to be as low as 0.57 V, which is much smaller than that of the referential PDI with 0.82 V. On the basis of highly sensitive action spectrum measurements on incident photon to current efficiency (IPCE), the above observation could be rationalized in terms of effective suppression of non-radiative recombination losses, which are attributable to a decreased LUMO-LUMO energy offset between the PTB7-Th and **2.4A** donor and the acceptor materials. As predicted by the computational studies based on the DFT calculations, it can be understood that the molecular design strategy for introducing multiple electron-donating groups at the bay positions would give rise to the lifted LUMO level to reduce the energy offset, offering beneficial effects to promote enhanced *V*_OC_, as seen in the case of the aforementioned **2.2A**. Finally, the work has successfully demonstrated that the progressive bromination/functionalization strategy will enable the tailor-made synthesis of new bay-functionalized perylene-based materials for exploring potentially efficient new NFA entries. It should be also emphasized that this strategy based on the PTE platforms offers greater synthetic and design flexibility to tolerate a variety of PDIs devoid of solubilizing groups at the imide positions.

## 3. Integrated Series of Perylene-Based Materials for Organic Photovoltaics

In the preceding chapter, the monomeric perylenes have been shown to present some inherent limitations in terms of inability to produce prominent photovoltaic performances with PCEs more than 10%, probably because of the difficulty of forming favorably organized structures of perylene chromophores, which will limit efficient carrier transportation in active layers [90]. Accordingly, the exploration of integrated molecular systems that consist of multiple PDI components may solve this problem due to the prospect of greater opportunities for achieving good crystallinity and charge-carrier mobility of the materials. In this regard, extensive research efforts have been initiated to develop the integrated series of perylene-based NFAs that motivate researchers to explore how structural shapes and dimensions affect photovoltaic performances. As a matter of fact, the synthetic approaches established for the monomeric PDIs have been applied to the synthesis of a diverse range of models categorized in this class of materials.

### 3.1. Bay-Linked Dimeric Perylene Diimides

Bay-linked dimeric PDIs, which consist of two PDI units mutually linked by covalent bonding at each bay position, are the simplest dimeric perylene-based materials. Synthesis of these compounds often employs Ullmann-type reductive homocoupling of bay-brominated perylene precursors. These dimeric PDIs are twisted around biaryl axes because of steric repulsions between adjacent bay-protons, which would be advantageous for suppressing excessive self-aggregation by breaking molecular planarity to promote good solubility, making them attractive platforms for potentially promising NFAs. As a representative research, Huo, Sun, Wang, and coworkers synthesized two types of bay-linked dimeric PDIs **3.1A** and **3.1B** with swallow-tail solubilizers at all the imide positions, one of which contains fused thiophene rings in both outer bay regions (Scheme 11) [91].

The synthetic elaboration involved preparative access to a bay-dichlorinated PDI monobromide **46** from a tetrachlorinated PDI **43**. The synthesis was initiated by dechlorination of the tetrachlorinated PDI **43** by using copper iodide and L-proline in heating dimethyl sulfoxide (DMSO) [92]. This allowed for selective removal of a pair of chlorine atoms placed in the same bay regions possibly through a bay-bridged Cu(III) complex **44** [93]. Exposure of the obtained dichloro PDI **45** to excessive bromine led to exclusive production of the monobromide **46**. In analogy with an unchlorinated PDI monobromide for preparing unfunctionalized bay-linked dimeric PDI **3.1A** [94], **46** underwent Ullman reaction to provide the corresponding dimeric PDI **47** with all the chlorine atoms remaining intact. The bridging sulfuration at the chlorinated bay positions by a palladium-catalyzed Stille-type coupling with bis(tributyltin)sulfide furnished the thiophene-fused dimeric PDI **3.1B**.

The dimeric PDIs **3.1A** and **3.1B**, which exhibited high solubilities because of the twisted molecular geometries and the solubilizing effects of the swallow-tails, were then blended with a polymeric donor PDBT-T1 at donor/acceptor ratios of 1:1 to fabricate the active layers via spin-casting from the CB solutions followed by thermal annealing. During the fabrication, use of less than 1% of DIO improved the cell performances as reflected by surface morphologies of the BHJ nanostructures, indicating that the PDI materials were allowed to aggregate properly and tuned to favorable phase separation. Indeed, under 1 sun illumination, the OSC based on **3.1A** produced an enhanced PCE of 5.40%, significantly higher than those of monomeric PDIs, with the following photovoltaic parameters: *V*_oc_ of 0.87 V, *J*_SC_ of 10.16 mA cm^−2^, and FF of 0.61. Meanwhile, **3.1B** achieved superior cell performance with PCE of 7.16%, *V*_oc_ of 0.90 V, *J*_SC_ of 11.98 mA cm^−2^, and FF of 0.66, which represent the highest levels of performances for NFAs reported at that time. The results were verified by the fact that the active layer exhibited high electron (*μ*_e_) and hole (*μ*_h_) mobilities of 2.8 × 10^−3^ and 1.2 × 10^−3^ cm^2^ V^−1^ s^−1^, respectively, resulting in nearly balanced charge transport of *μ*_e_/*μ*_h_ = 2.3, as observed by the space charge-limited current (SCLC) measurements. One of the reasons for the superior performance of **3.1B** can be rationalized on the basis of lifted LUMO level arising from electron-donating nature of the bay-bridging sulfur atoms, which is potentially advantageous for raising *V*_oc_ to deliver the higher level of PCE. The work on the dimeric PDIs has demonstrated considerable potential of the integrated perylene-based NFAs to yield encouraging device performances [95,96,97,98,99,100].

### 3.2. Bay-Bridged Dimeric Perylene Diimides

Alternatively, bay-bridged dimeric PDIs, which possess a functional spacer between two PDI units, seem to be an attractive series of novel NFA materials. In designing the bay-bridged systems, insertion of bulky spacers into the central cores of molecular backbones may lead to an efficient shielding of overstrong π–π interactions of the planar perylene nuclei to circumvent serious insolubility problems of the PDI materials. As a noteworthy attempt, Pei and Zhao employed spirobifluorene as a spacer to generate a spirobifluorene-bridged dimeric PDI **3.2A** denoted commonly as SF-PDI_2_, which represents a potentially promising NFA candidate commercially available from Sigma-Aldrich Co. Inc. [101]. The synthesis was accomplished by a palladium-catalyzed cross-coupling between PDI monobromide **48** and 9,9-spirobifluorene-2,7-diboronic acid bis(pinacol) ester (Scheme 12). In an initial investigation for the OPV application, **3.2A** was blended with the polymeric donor P3HT at a donor/acceptor ratio of 1:1. With respect to morphological feature, AFM images of the active layer showed a smooth surface because of a high degree of miscibility in the BHJ blend. The OSC based on **3.2A** produced a PCE of 2.35% under 1 sun illumination with the following photovoltaic parameters: *V*_oc_ of 0.61 V, *J*_SC_ of 5.92 mA cm^−2^, and FF of 0.65. Electrochemical analysis indicated that the LUMO and HOMO levels of **3.2A** were slightly lifted but substantially unaffected when compared to that of the monomeric PDI, providing valuable information that the spirobifluorene spacer should have little influence on FMO levels of the PDI units.

In another work, **3.2A** has been applied for more efficient OPV devices by blending with a newly designed polymeric donor P3TEA developed by Gundogdu, Gao, Yan, and coworkers [102]. In practice, the OPV devices fabricated using a blend at a donor/acceptor ratio of 1:1.5 were shown to reproduce an impressive average PCE of 9.13% for 30 devices under 1 sun illumination, giving the average of the following photovoltaic parameters: *V*_oc_ of 1.11 V, *J*_SC_ of 13.15 mA cm^−2^, and FF of 0.627. Notably, it can be seen that the highest PCE estimate reached as high as 9.47% with the respective parameters; *V*_oc_ of 1.11 V, *J*_SC_ of 13.27 mA cm^−2^, and FF of 0.643. In this respect, the work emphasized the enhanced *V*_oc_ representing the largest level found for the state-of-the-art OSCs. The researchers made intense theoretical efforts to gain detailed insights into the origin of the observed large *V*_oc_ quantity. By probing the rate and driving force for charge separation, the authors conducted thin film spectrophotometric measurements of the Fourier-transform photocurrent spectroscopy external quantum efficiency (FTPS-EQE) and electroluminescence (EL) spectra. These experiments revealed small *V*_loss_ of as low as 0.61 V and allowed the estimation of two constituent factors, radiative recombination loss originating from the absorption below the band gap (*V*_OC_^rad,below gap^) of 0.07 V and non-radiative recombination loss (*V*_OC_^nonrad^) of 0.26 V, which are comparably low to those of the inorganic devices such as *c*-Si and perovskite solar cells. In this situation, the energy of the CT state (*E*_CT_) is very close to that of singlet exciton of the lower band gap donor component in the blend (*E*_gap_), which presented a conclusive picture that the driving force for charge separation (*E*_gap_*-E*_CT_) is extremely small or nearly negligible. The work successfully demonstrated that a small driving force can maximize the achievable efficiency of OSCs by reduction of *V*_loss_ due to the non-radiative recombination processes, which will expand the potential possibilities of NFA-based OPV systems to lead toward a more practical technology. Although numerous examples of the bay-bridged dimeric PDIs have been explored so far [103,104,105,106,107,108,109,110,111,112,113,114,115,116,117,118,119], to the best of our knowledge, the above mentioned **3.2A**/P3TEA system exhibited the highest performance due to the photovoltaic properties with the minimal voltage loss and the drastically enhanced *J*_SC_.

### 3.3. Bay-Fused Dimeric and Tetrameric Perylene Diimides

According to the reports on the bay-linked and bay-bridged dimeric PDIs, nonplanar perylene-based materials have beneficial effects on suppression of undesirable phase separation from the donor materials, leading to preferable morphological properties to achieve enhanced cell performances. Nevertheless, local flexibility due to unrestricted rotation around the inter-ring C-C bonds will induce detrimental effects on molecular packing and bulk crystallinity in thin films, which might result in some disadvantageous situations to degrade cell performances. Thus, it is desirable to search for alternative structures with restricted conformation. In this context, Jian, Wang, and coworkers have synthesized a unique class of the integrated PDI systems **3.3A** and **3.3C**, which consist of the two PDI units connected together to the same carbon atom to adopt an orthogonally twisted arrangement in a spirocyclic form (Scheme 13) [120].

This structural system was constructed by a reaction based on a palladium-catalyzed intramolecular dual alkylation developed by Zhang and coworkers, which enables incorporation of methylene groups between two aromatic rings by use of dibromomethane through C-H and C-X functionalization [121]. Application of this methodology to the monobrominated PDI **49** has unexpectedly provided spiro-fused dimeric PDI **3.3A** in high yield of 81%. This intermolecular reaction, which is different from the precedent literature, could be explained as follows. First, oxidative addition of the PDI bromide to Pd(0) and subsequent C-H metalation formed palladium-bridged PDI. Then, predominant two-fold oxidative addition of the methylene source produced the dimeric PDI **50**. After reductive elimination providing methylene-bridged PDI dimer **51**, further C-H-metalations and reductive eliminations might give **3.3A** through a cyclometalated intermediate **52**. The X-ray crystallography of **3.3A** revealed the orthogonal orientation of the two PDI subunits that are arranged in very close contact with the perylene planes of adjacent π-stacks. Steady-state absorption characteristics of the solution formed in CF showed the presence of interchromophore interactions within the molecular framework, which gave rise to significant bathochromic shift of the absorption maximum by up to 23 nm relative to that of monomeric PDI, providing an energy gap (*E*_g_) estimate of 2.17 eV.

Extension of the synthetic procedure by replacing dibromomethane with 1,1,2,2-tetrabromoethane led to a formation of an ethylene-bridged dimeric PDI **3.3B**, essentially identical with that obtained by the conventional approach of Xiao, Ng, Nuckolls, and coworkers [122]. The palladium-catalyzed spirocyclization reaction could also be applied to a monobrominated ethylene-bridged dimeric PDI **53** which was prepared by the conventional bromination of **3.3B** with a moderate yield of 40%. Under the catalytic conditions with dibromomethane, **53** was transformed into spiro-fused tetrameric PDI **3.3C** that contains the two ethylene-bridged dimeric PDI units. The tetramer **3.3C** was typically characterized by broad absorption features ranging from the ultraviolet to visible regions to give the *E*_g_ estimate of 2.18 eV, which can be mostly regarded to be superimposed by the absorption profiles of the two constituent subunits, namely **3.3A** and **3.3B**.

The compounds **3.3A**, **3.3B**, and **3.3C** were applied as NFA materials to fabricate the respective BHJ blends spin-coated with the polymeric donor PTB7-Th at donor/acceptor ratios of 1:1 in CB solutions containing the solvent additive DIO. The analysis of the photovoltaic systems under 1 sun illumination displayed impressive PCE values of 5.18%, 6.43%, and 7.18% for **3.3A**, **3.3B**, and **3.3C** with the following photovoltaic parameters: *V*_oc_ of 0.83, 0.80, and 0.77 V; *J*_SC_ of 12.67, 13.50, and 14.58 mA cm^−2^; FF of 0.47, 0.56, and 0.60, respectively. Thus, **3.3C** was proven to be apparently superior to the others with regard to photovoltaic activity. In agreement with expectations from the use of chemically unmodified PDI platforms, all the OSCs were shown to yield comparable *V*_oc_ values. This work has clearly demonstrated that rational design of SMA systems integrating multiple PDI chromophore units provides effective ways for achieving higher photovoltaic performances.

### 3.4. Propeller-Shaped Trimeric and Tetrameric Perylene Diimides

It is clear now that the designed perylene-based materials, comprising multiple PDI units with twisted geometries, are suitable NFAs for further applications toward high-performance thin-film photovoltaic technology. As a further elaborated design concept, covalently linked trimeric or tetrameric PDI systems, which could adopt propeller-like geometries, are expected to offer superior material properties compared to the structurally simpler monomeric and dimeric PDIs due to the spatially extended arrays of perylene nuclei, ensuring efficient intermolecular π–π contacts between the donor and acceptor materials. In practice, this class of compounds is accessible through bay-brominations and catalytic cross-coupling reactions by exploiting spacer building blocks with three or more functional groups as coupling partners.

As a representative example, Liu, Shi, Chen, and coworkers have developed a propeller-shaped trimeric PDI **3.4A** that consisted of a 1,3,5-trisubstituted benzene core and three PDI units [123]. A palladium-catalyzed cross-coupling reaction of the PDI monobromide **48** with commercially available benzene-1,3,5-triboronic acid bis(pinacol) ester easily provided **3.4A** in 86% yield (Scheme 14). This compound exhibited a good film-forming ability as characterized by thermal analyses, indicative of suppressed crystallinity due to the twisted geometries of the three inter-ring linkages. This feature led to good miscibility with the polymeric donor PTB7-Th, as confirmed by grazing incidence X-ray diffraction (GIXD) studies. Systematic exploration of the preparative conditions at various donor/acceptor ratios led to the optimum results obtained in the photovoltaic characterization when the donor/acceptor ratio was set as 1:1.5. Additionally, use of a small amount of CN increased the cell performances to achieve a PCE of 5.65% under 1 sun illumination with the following photovoltaic parameters: *V*_oc_ of 0.83 V, *J*_SC_ of 13.12 mA cm^−2^, and FF of 0.52, whereas DIO was ineffective. The impact of the CN additive was visualized by SCLC to observe an approximately two-fold increase of the carrier mobilities with *μ*_e_ and *μ*_h_ of up to 4.20 × 10^−5^ and 17.48 × 10^−5^ cm^2^ V^−1^ s^−1^ for the BHJ blends, respectively. The work has successfully demonstrated that the propeller-shaped trimeric systems can promise superior performances for the OPV applications.

At almost the same time, Li, Sun, Wang, and coworkers independently developed another type of propeller-shaped trimeric PDIs **3.4B** and **3.4C** [124], structurally quite analogous to **3.4A** (Scheme 15). In their work, the trimeric PDI **54** was subjected to oxidative photocyclization with iodine, which efficiently proceeded under irradiation with a 500 W mercury lamp in toluene to furnish the fully ring-fused aromatic system **3.4B** in 93% yield. The ring-fused system has more conformational rigidity due to restricted rotation around the inter-ring C-C bonds between the core and PDI units. Further elaboration to an additional candidate was carried out by transformation to the corresponding selenophene-fused trimeric PDI **3.4C**. Nitration of **3.4B** followed by selenylation of **55** at 190 °C with selenium powder afforded **3.4C** in excellent yields. Of note, **3.4B** and **3.4C** exhibited enhanced solubilities in common organic solvents such as DCM, CF, and *o*-dichlorobenzene (DCB), which were mainly conferred by incompletely planarized geometry due to the large steric repulsion among the PDI units as revealed by X-ray crystallography. Absorption characteristics of these compounds were found distinguishable as seen by bathochromic shift of up to 12 nm for **3.4C**, which reflects the tuned energy levels of FMOs with slightly lifted LUMO level presumably caused by the bay-selenylations.

The trimeric PDIs **3.4B** and **3.4C** were applied to photovoltaic characterization by fabrication of BHJ blends with the polymeric donor PDBT-T1 with the donor/acceptor ratios of 1:1, identified as an optimal stoichiometric balance for both devices. Spin-casting of the blend solutions in DCB which contain tiny quantities of DIO gave higher quality BHJ films with quite smooth and uniform surfaces compared with those without the additive as observed by AFM. The OSCs employing the respective active layers produced remarkably high PCEs of 8.28 and 9.28% with the following photovoltaic parameters: *V*_oc_ of 0.97 and 1.00 V, *J*_SC_ of 12.01 and 12.53 mA cm^−2^, and FF of 0.701 and 0.717 for the best records of **3.4B** and **3.4C**, respectively. The particularly high *J*_SC_ and FF estimates were verified by the SCLC studies to measure the charge-carrier mobilities of the optimal blends. Indeed, the analyses showed very high values with *μ*_e_ of 1.5 × 10^−3^ and 2.2 × 10^−3^ cm^2^ V^−1^ s^−1^ and *μ*_h_ of 1.0 × 10^−3^ and 1.7 × 10^−3^ cm^2^ V^−1^ s^−1^, leading to good balanced charge transport properties with *μ*_e_/*μ*_h_ = 1.5 and 1.3 for **3.4B** and **3.4C**, respectively. Intriguingly, the selenophene-fused **3.4C** highlights a noticeable impact of the electron-donating elements on the improved *V*_oc_, which can be attributed to lifted LUMO levels of the selenylated PDI units. This work has successfully demonstrated that the propeller-shaped systems possessing conformationally locked and distorted structures offer potential advantages with regard to generation of high morphological order and enhanced optoelectronic properties to achieve the next higher levels of device performances.

As a more sophisticated system of the oligomeric PDIs, Yan and coworkers designed PDI tetramers **3.4D** and **3.4E** possessing a tetrathienylbenzene (TTB) linker (Scheme 16) [125]. The TTB has been developed as a hole-transporting material by de la Torre, Nazeeruddin, Torres, and coworkers [126], as a potential option for the core building block. The synthesis started from a palladium-catalyzed cross-coupling between 1,2,4,5-tetrabromobenzene (**56**) and 2-(tributylstannyl)thiophene with bis(dibenzylideneacetone)palladium (Pd(dba)_2_) and tri(*o*-tolyl)phosphine in refluxing toluene to give a 87% yield of **57**, which was then subjected to an iridium-catalyzed C-H borylation using (1,5-cyclooctadiene)(methoxy)iridium dimer ({Ir(OCH_3_)cod}_2_) and 4,4′-di-*tert*-butyl-2,2′-dipyridyl (dtbpy) under microwave irradiation in heptane at 120 °C to furnish tetraboronic ester **58** in 77% yield. Finally, palladium-catalyzed cross-coupling of **58** with the PDI monobromide **48** using tris(dibenzylideneacetone)dipalladium (Pd_2_(dba)_3_) and 2-dicyclohexylphosphino-2′,6′-dimethoxybiphenyl (SPhos) under microwave irradiation in aqueous THF at 80 °C led to generate TTB-cored tetrameric PDI **3.4D** in 62% yield. This compound was further converted to the fused derivative **3.4E** by an oxidative cyclization using FeCl_3_ and nitromethane in toluene in 92% yield [127]. The ring fusion resulted in significant enhancement of photon absorptivity and large blue-shift in absorption peaks. Moreover, the ring co-planarization in **3.4E** dramatically alters the conformational preference to adopt a “double-decker”-shaped geometry with each pair of two PDI units stacked homogeneously rather than “propeller”-shaped structure of **3.4D**, as predicted by DFT calculations.

The potential utility of **3.4D** and **3.4E** was demonstrated by measuring photovoltaic properties of the respective BHJ active layers, being fabricated by spin-coating the blend solutions containing the polymeric donor P3TEA and tiny amount of a solvent additive 1,8-octanedithiol (ODT) in 1,2,4-trimethylbenzene (TMB) at the donor/acceptor ratios of 1:1.5 and the following thermal annealing. Indeed, under 1 sun illumination, both OSCs displayed remarkably high average PCEs of 6.67 and 10.37%, the best of which reached 7.11 and 10.58% for **3.4D** and **3.4E**, respectively. The corresponding average photovoltaic parameters were given as *V*_oc_ of 1.05 and 1.13 V, *J*_SC_ of 12.06 and 13.89 mA cm^−2^, and FF of 0.529 and 0.659 for **3.4D** and **3.4E**, respectively. According to the equation of subtracting *V*_oc_ from *E*_g_/*q* for the BHJ blend, where *q* is the elemental charge, *V*_loss_ of the ring-fused PDI was calculated to be 0.53 V, representing the lowest level for OSCs at the time [128,129]. The low *V*_loss_ was attributed to the higher LUMO level of **3.4E**. The OPV device based on **3.4E** also showed higher internal (IQE) and external quantum efficiencies (EQE) of 80 and 68% than **3.4D** (IQE of 64 and EQE of 55%). This fact was ascribed to the strong absorption of the shorter wavelength region from 350 to 500 nm. Besides, the observed higher *J*_SC_ and FF of the device based on **3.4E** were explained as a consequence of several enhancement factors, such as high light absorption capability, better energetic and absorption match of the SMA with the donor material because of the lifted LUMO level, and improved molecular packing of the active layer due to beneficial impact of the “double-decker” geometry of the SMA on enhancing charge transport ability. In fact, the BHJ blend fabricated with **3.4E** exhibited SCLC mobilities of *μ*_e_ = 1.1 × 10^−4^ and *μ*_h_ = 1.5 × 10^−4^ with the electron and hole current balance of *μ*_h_/*μ*_e_ = 1.4, more preferable to help induce high *J*_SC_ and FF rather than those for **3.4D** (*μ*_e_ = 0.37 × 10^−4^, *μ*_h_ = 1.1 × 10^−4^, and *μ*_h_/*μ*_e_ = 3.0). This work has successfully demonstrated that a remarkable milestone yielding PCE beyond 10% can be realized in OSCs based on perylene-based NFAs as well as the rationally designed materials of this class have the potential to exceed device performances achieved in fullerene-based OSC.

Recently, the PCE record established in the above PDI system has been exceeded by another type of tetrahedral triarylphosphine-cored trimeric PDI system **3.4F** reported by Ma, Peng, and coworkers (Scheme 17) [130]. The most noteworthy attempt in this work is the introduction of a new solvent additive 4,4′-biphenol (BPO) that leads to improved crystallinity of active layers via enhancing supramolecular interactions between the donor and acceptor materials. This new post-treatment method was termed as “molecular lock” concept.

The core building blocks of triarylphosphine tribromides **59** and **60** were readily available from commercial reagents according to the literature [131] and the following oxidation and sulfuration of phosphine with hydrogen peroxide and sulfur, respectively. As a PDI reagent, pinacolatoboron-substituted PDI **62** was prepared from the relevant PDI monobromide **61** by Miyaura borylation using [1,1′-bis(diphenylphosphino)ferrocene]dichloropalladium(II) (PdCl_2_(dppf)) [132] with a satisfactorily yield of 78%. The cross-coupling of **59** and **60** with **62** under the standard conditions using tetrakis(triphenylphosphine)palladium(0) (Pd(PPh_3_)_4_) gave the trimeric PDIs **3.4F**, **3.4G**, and **3.4H** in 42–46% yields. All the three compounds exhibited very similar absorption characteristics, indicative of little interference exerted by the electronic structures of the phosphine cores. These observations were consistent with comparable electrochemical band gaps of 2.18, 2.21, and 2.15 eV for **3.4F**, **3.4G**, and **3.4H**, respectively. The BHJ blends were prepared from the respective materials and the polymeric donor PTTEA, which can be substantially recognized as an architectural congener of P3TEA with a nearly identical structure, in TMB at the donor/acceptor ratios of 1:1.5. The fabrication of the active layer was then performed by spin-coating the blend solutions that contain 3 wt % of BPO to obtain high-quality films as confirmed by morphology analyses such as AFM and TEM. It is of note that the OPV devices as prepared with the BPO additive produced remarkably high PCE estimates of 11.01, 6.19, and 9.67% under 1 sun illumination with the following photovoltaic parameters: *V*_oc_ of 0.99, 1.02, and 0.97 V, *J*_SC_ of 14.89, 12.25, and 14.20 mA cm^−2^, and FF of 0.747, 0.495, and 0.702 for the best records given by **3.4F**, **3.4G**, and **3.4H**, respectively. In comparison with cell performances obtained from BPO-free OSCs, these PCEs were raised by 2.36, 1.01, and 1.67% for the devices with **3.4F**, **3.4G**, and **3.4H**, respectively. This PCE enhancement was ascribed to the joint “molecular lock” effect. Namely, supramolecular hydrogen bonding networks via O-H···F and O-H···O=P interactions trigger morphological changes in crystallinity and interpenetrating nanostructures of the BHJ active layers. The work has for the first time exemplified the success of perylene-based OSC on achieving PCE over 11% and successfully demonstrated that the use of greener solvent additive BPO was validated as an effective method to enhance photovoltaic and morphological properties of the OPV devices. In agreement with many other preceding molecular systems of this class [133,134,135,136,137,138,139,140,141,142,143,144,145,146,147,148,149,150,151], the results given in the work strongly assert that the propeller-shaped PDI systems will provide a promising platform to achieve high performances for OPV applications.

## 4. Conclusions

There have been a number of successful attempts to develop perylene-based NFAs that can provide potential activity in photovoltaic systems to promote efficient light to electric power conversion as viable substitutes for fullerene acceptors when coupled with suitable donor materials. The present review has explored some representative achievements in the developments of rationally designed PDI systems, which highlight the synthetic methodology for creating well-designed molecular nanostructures and the structure-performance relationship of the SMAs for the photovoltaic outcomes. Table 1 summarizes selected data of the photovoltaic records achieved by the PDI systems that are presented in this review. According to the literature data, schematic energy level diagrams of the donor and PDI materials used in the OPV devices are represented in Figure 6.

A survey of the relationship between the molecular geometry and the device characteristics gives several indications with respect to the promising design of perylene-based NFAs, which are represented as follows. (1) Deformation of planarity in molecular backbones of the PDI materials is effective to ensure high solubility. (2) Structures with restricted conformational flexibilities are advantageous to enhance the crystallinity of active layers. (3) Avoidance of flexible long alkyl chains such as swallow-tail solubilizers will provide an efficient strategy to reduce the *V*_loss_ due to non-radiative deactivation. (4) Integrated molecular systems that consist of multiple PDI units are preferred to increase electron mobilities resulting in significant enhancement of high *J*_SC_ and FF estimates. (5) The introduction of electron-donating groups on the PDI core has significant impact in terms of lifting LUMO levels to improve *V*_OC_ as a consequence of reduced energy offset. (6) Control of an intermolecular network via non-covalent interactions can lead to an altered surface structure to achieve better crystallinity and interpenetrating nanostructures of BHJ layers. (7) Low-energy absorptivity of molecules that harvest near infrared (NIR) solar photons is desirable to find new opportunities for fabricating novel perylene-based NFAs with high potential. These considerations will provide practical guidelines in rational design of new perylene systems for spawning future generations of OSCs. One option considered is to use the polymeric systems that contain a large number of PDI units in a unimolecular structure, but attempts made with such polymer materials fail to achieve remarkable success in manifestation of prominent cell performances [152,153,154,155,156]. Hence, it is reasonable to expect that oligomeric materials would offer highly promising options and exploratory investigations on such substances will be the next challenge in the future developments of perylene-based NFAs.

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
