# Peer review of "Development of Perylene-Based Non-Fullerene Acceptors through Bay-Functionalization Strategy"

_materials, 2020, doi:10.3390/ma13092148_

Round 1

Reviewer 1 Report

This MS is well written and references are pretty well selected. However, I would suggest to the authors instead of "simply" citing a package of references to build some table with structure of the PDI derivatives, donor partner, Voc, FF, Jsc and PCE. For instance, authors wrote :"The work on the dimeric PDIs has demonstrated considerable potential of the 539 integrated perylene-based NFAs to yield encouraging device performances [80-85]." So, a simple table with structures, PV characteristics and reference would clearly level up the quality of the paper. Finally, I also would suggest the authors to discuss this paper : https://www.nature.com/articles/s41598-020-60012-7 In the submitted review, all exemples concern one isomers. Should be good to open the conclusion on the results from this paper showing that in some cases, 1,6 isomer can lead to better performances than the usually used 1,7. Anyway, this review is easy to read and should be considered for publication after addressing these few comments.

Author Response

Answers for the comments by the reviewer

  1. The reviewer pointed out that the authors should discuss the recent work on the difference in photovoltaic properties between 1,6- and 1,7-regioisomeric PDIs.

              According to the comment, we inserted new sentences "Conventionally, 1,6-difunctionalized PDIs derived from 1,6-dibromides have been thought to be unwanted side products that need to be removed for use in photovoltaic systems. However, it is interesting at this point that recent work has demonstrated the superior photovoltaic properties of 1,6-difunctionalized PDI to its 1,7-isomer, contrary to expectations that there is a negligible influence of the difference in substitution patterns of the PDI systems on the material properties [65]." at the relevant place of the text (page#7, line#254 in the latest revised manuscript) and added a new reference#65 as suggested by the reviewer.

Reviewer 2 Report

The manuscript submitted to Materials entitled “Development of Perylene-based Non-Fullerene Acceptors” by Takahashi and co-workers presents a review of acceptor materials for organic photovoltaic (OPV) devices that are based on perylene diimide and related structures. The article provides an introduction to perylenes and organic photovoltaics before discussing the synthesis and application of different types of perylene-based materials. These include monomeric perylenes, bay-difunctionalised perylene diimides and propeller-like structures. The limitations and advantages of the different structure types are discussed as a means of establishing design rules for perylene-based materials. The discussions on the functionalisation of the perylene units is very important and useful for researchers who may look to design new perylene-based non-fullerene acceptors. Overall, I believe the article is detailed and well-presented. By combining discussions on the limitations and successful synthetic routes for perylene derivatives and the effect of different structure types on photovoltaic performance I believe the authors have written a review article which will be a useful reference for researchers working in organic semiconductor field, whether they focus on organic synthesis or device work.

There are a few points that should be addressed to improve the manuscript prior to publication:

  • The authors discuss examples of monomeric perylene structures giving reasonably low power conversion efficiencies when used in OPV devices. This is deemed to be due to “due to difficulty in forming favourable organized structures of perylene chromophores”. It would be helpful if the authors could present a more detailed comparison with experimental evidence of this by reporting the electron mobilities of the materials or studies of the film topographies, for example.
  • The authors also state that “Numerous examples of bay-bridged PDIs have been explored so far” and use multiple citations to indicate this but focus mainly on material 3.2A for the discussion of 3.2A. It would be helpful if the authors expanded on some of the cited examples of bay-bridged PDIs and highlighted the effect of structural variation on photovoltaic performance. What is it that makes 3.2A a good acceptor compared to other bay-bridged PDIs?
  • Similarly, the article presents a clear evolution of the structures of non-fullerene acceptors based on perylene-based materials. However, it would be useful if the authors could provide a critical comparison of perylene-based materials and non-fullerene acceptors without perylene. What advantages/disadvantages to using perylene-based materials and can these be adapted to match or improve OPV performance in the highest performing Donor:NFA blends reported?
  • The presentation of different structures in the middle of 2 different reaction schemes in figure 1 makes this figure unclear. It would be useful to separate example structures away from the reactions presented. Additionally, it seems unusual to present the synthesis of PTCBI as a representative example of chemistry that can be carried out when there is very little discussion on such materials.
  • There are some minor typos/errors that should be corrected prior to publication. These include:

Figure 1 – change pery to peri

Line 81 and figure 3 – change PC60BM to PC61BM

Line 145 – change ortho (2,5,8,12-) to (2,5,8,11-)

Line 281 – Intersystem crossing more commonly referred to as ISC (not IST)

Scheme 6 – last arrow thicker than the others

Line 425 – Define acronym AN (presumably acetonitrile)

In summary, the manuscript presents a clear discussion on the challenges in the synthesis of perylene-based acceptor materials and how structural variation can impact on organic photovoltaic performance. This review will be a useful guide for researchers in the OPV community, particularly as non-fullerene acceptors have now been used in the most efficient OPV devices reported. Therefore, I believe this article can be accepted for publication in Materials.

Author Response

Answers for the comments by the reviewer #1

  1. The reviewer pointed out that the authors should present experimental evidence of reporting the electron mobilities with regard to the issue of low PCE for monomeric PDIs.

              According to the comment, we added a new reference#89 at the end of the related sentence "chromophores, which will limit efficient carrier transportation in active layers" (page#13, line#481 in the previous manuscript) to present comparison of experimental evidence including the electron mobilities.

  1. The reviewer pointed out that the authors should explain why the 3.2A/P3TEA system is highlighted.

              According to the comment, we added a new description "due to the photovoltaic properties with the minimal voltage loss and the drastically enhanced JSC" at the end of the related sentence "Although numerous examples of the bay-bridged dimeric PDIs have been explored so far [88-104], to the best of our knowledge, the above mentioned 3.2A/P3TEA system exhibited the highest performance" (page#15, line#574 in the previous manuscript).

  1. The reviewer pointed out that the authors should give some comments on comparison of perylene-based NFA with perylene-free NFA.

              According to the comment, we added a new description "and suffer from synthetic complexity [32]" at the end of the related sentences " However, these types of NFA materials still face performance degradation due to their susceptibility to moisture and oxygen, offering less long-term durability [30,31]" (page#4, line#132 in the previous manuscript).

  1. The reviewer pointed out that the authors should make appropriate revisions on the arrangement of the chemical structures presented in Figure 1.

              According to the comment, we made appropriate revisions of Figure 1 so as to obtain a description of synthetic accessibility to various types of perylene derivatives (page#1, line#32 in the previous manuscript).

  1. The reviewer pointed out that the authors should correct typos/errors found in the manuscript.

              According to the comment, we corrected all the typos/errors indicated by reviewers as follows.

(1) "pery" given in Figure 1 (page#1, line#32 in the previous manuscript) was replaced with "peri".

(2) All "PC60BM" (page#3, line#87 and Figure 3 (page#3, line#97) in the previous manuscript) was replaced with " PC61BM ".

(3) "ortho (2,5,8,12-)" (page#4, line#142 in the previous manuscript) was replaced with "ortho (2,5,8,11-)".

(4) All "IST" (page#7, line#275 and page#8, line#302 in the previous manuscript) were replaced with "ISC".

(5) Thickness of the last arrow given in Scheme 6 (page#10, line#363 in the previous manuscript) was adjusted.

(6) Acronym of AN was defined by replacing the relevant description "heating AN" (page#11, line#418 in the previous manuscript) with "heating acetonitrile (AN)".

Reviewer 3 Report

The manuscript "Development of Perylene-Based Non-Fullerene Acceptors" by Keisuke Fujimoto, Masaki Takahashi, Seiichiro Izawa, and Masahiro Hiramoto gives an overview of the recent progress that has been made in the field of perylene based organic solar cells. In general, this review article is well structured, straight forward to read, and does not get lost in unnecessary details, which often happens in other review articles (specifically choosing representative compounds to convey core concepts is a good strategy in my opinion). Although there have been numerous other articles of a similar topic, most of them were written nearly a decade ago, which underlines the need for a fresh look into this group of materials, which recently has been somewhat overshadowed by the various types of very high-performing, thiophene-based NFAs. Before publication may happen, there are a few points that warrant some revision:

1. The introduction to OPVs could be improved a bit more by discussing further selling points of this technology such as semi-transparency (discussed in ref. 16 and also here: 10.1021/acsenergylett.9b00721). Furthermore, it should be highlighted more how OPV PCE was on a plateau of ca. 10% when using fullerenes, which was specifically a problem during the times of the discovery of perovskite solar cells and their rapid PCE increases beyond 15%. Only since the adoption of high performing NFAs are more research groups focusing on OPV again (this can be easily be observed in the current NREL chart: https://www.nrel.gov/pv/assets/pdfs/best-research-cell-efficiencies.20200311.pdf, or Green's solar cell efficiency tables: 10.1002/pip.3228). An additional reference focusing more on the economics of OPV is also Seth Darling's paper published in 2013 (10.1039/C3RA42989J). Furthermore, in the following article PCEs approach 18% for single junction BHJ solar cells were described, although they "only" reached a certified PCE of 17.3% (10.1002/adma.201908205).

2. I would also suggest to reference in the introduction more previous review articles and papers that focused on perylenes. It is true that most of them were published at the beginning of the last decade, but it would still be prudent to add these important resources to the current manuscript as it puts it into a scientific/historical context. The following articles should be added:
a) The Rylene Colorant Family (10.1002/anie.200902532).
b) Bay-extension of PDIs (10.1021/ol201623f).
c) Review on enlarged PDIs (also includes OPV, although only very limited) (10.1039/C000137F).
d) Perylenes for OPVs (also quite old paper, but still important today) (10.1002/adma.201104447).
e) Perylne esters as an alternative for PDIs (up to now this never really took off, but in my opinion it should be added for a complete picture) (10.1039/C3PY00938F).

3. On page 4, the authors claim that extensions in the ortho- and bay-position(s) are two distinct classes of PDI extensions. I would suggest to also mention the extension at the peri-position, which was extensively studied by Müllen and coworkers. Technically these extended compounds at the peri position are not perylenes anymore, but this is only a minor issue.

4. On page 5, the authors describe the addition of swallow tailed side chains as an important strategy that enhances the solubility of PDIs. This is true, but an alternative strategy is to operate with PTEs / perylene esters during synthesis and as a final step conduct saponification and amination. In scheme 1 of the following paper (10.1002/chem.201403287) this is exemplified. It is noteworthy that the saponification and amination have high yields of 98% and 97% in this example! Of course, this strategy is also not perfect as it is limited to only some types of bay-extensions (certain functional groups on the extensions might not survive the harsh conditions of saponification/amination).

5. On page 5 line 190, the authors describe that the focus of their manuscript lies on PDIs extended/functionalized at the bay position. This is a valid strategy to ensure that the manuscript is succinct and focused, but this should have been made clear from the beginning (abstract) and should be reflected also in the title. Therefore, I suggest to extend the title in such a way that the main focus of the paper is reflected already in the title.

6. On page 6, line 239, the authors highlight the effect of core deformation on solubility. I would like to add that this has been also discussed in the framework of reducing the transition temperature in liquid crystalline/mesogenic perylene derivatives (10.1002/anie.201108886; 10.1080/15421406.2017.1284387). In general, solubility and mesogenic properties are somewhat linked, i.e. the reduction of the transition temperature will most likely also result in better solubility. Most of the time the use of liquid crystalline perylenes was investigated in the framework of OLEDs, though (10.1021/acsami.5b03496). But this was also mentioned in ref. 43 of this manuscript and could maybe shortly be discussed for OPV as one interesting strategy offered by functionalized perylenes.

7. On page 13, the authors discuss the HOMO and LUMO levels of some PDI derivatives. This manuscript would greatly improve, if at least the HOMO and LUMO energies of the compounds (acceptors and donors) listed in table 1 would be depicted in a separate figure. This would allow the reader to quickly see the relevant energy levels and potential advantages of the listed reference compounds.

8. On page 19, line 730 it is claimed that a Vloss of 0.53 V was achieved, being the lowest one reported. This might have been correct at that time, but there are some examples where Vloss below 0.53 V is reported (10.1002/aenm.201701073; 10.1021/acsenergylett.0c00029).

9. Minor comments, typos, and grammatical errors can be found in the attached manuscript that includes reviewer comments.

Author Response

Answers for the comments by the reviewer #2

  1. The reviewer pointed out that the authors should improve the introduction to OPVs by discussing selling points of their technology such as semi-transparency.

              According to the comment, we inserted a sentence "Furthermore, the technological applications of the solution processing methods provide attractive options to fabricate semitransparent devices that allow for harnessing solar power through colored windows and roofs of buildings and vehicles [21]." at the relevant place of the text (page#3, line#103 in the previous manuscript) and added a new reference#21 as suggested by the reviewer. Furthermore, we added a new reference#24 to underline the economic advantages of OPVs as suggested by the reviewer, which accompanies the need for addition of the relevant citation by replacing "[20]" (page#3, line#110 in the previous manuscript) with "[23,24]". In a similar way, we inserted a sentence "Further research efforts with a Y6 derivative have made the most successful achievement in this field with remarkably enhanced PCE approaching 18% [34]." at the relevant place of the text (page#4, line#132 in the previous manuscript) and added a new reference#34 to feature an enhanced capability of the state-of-the-art OSC as suggested by the reviewer.

  1. The reviewer pointed out that the authors should add important resources to enrich reference entity records.

              According to the comment, we added the following new references as suggested by reviewer.

(a) A new reference#44 was added to underline the importance of the rylene colorant family, which accompanies the need for addition of the relevant citation by replacing "[36]" (page#4, line#153 in the previous manuscript) with "[43,44]"

(b) A new reference#57 was added to underline the importance of the bay-extension of PDIs, which accompanies the need for addition of the relevant citation at the end of the related sentence "on the perylene-based SMAs developed through bay-functionalization strategies." (page#5, line#187 in the previous manuscript).

(c) A new reference#40 was added to feature the enlarged PDIs, which accompanies the need for addition ofthe relevant citation at the end of the related sentence "structural diversity to offer a broad range of functional properties." (page#4, line#141 in the previous manuscript).

(d) A new reference#11 was added to underline the importance of perylenes for OPVs, which accompanies the need for addition of the relevant citation at the end of the related sentence "perylene dyes as important constituents in organic photovoltaics (OPVs)." (page#2, line#54 in the previous manuscript).

(e) A new reference#39 was added to feature the perylene esters as an alternative for PDIs, which accompanies the need for addition of the relevant citation by replacing "[32,33]" (page#4, line#138 in the previous manuscript) with "[37-39]".

  1. The reviewer pointed out that the authors are recommended to mention the research works on the extension at the peri-positions.

              According to the comment, we added a new reference#2 to feature the research work by Müllen and coworkers on the extension at the peri-positions of perylene dyes as suggested by the reviewer, which accompanies the need for addition of the relevant citation at the end of the related sentence "withdrawing elements into the aromatic nucleus."(page#1, line#31 in the previous manuscript).

  1. The reviewer pointed out that the authors should mention an alternative strategy involving the use of PTEs and their saponification/imidation to access PDIs.

              According to the comment, we inserted a sentence "In this regard, operating with more soluble perylene tetracarboxylic esters (PTEs) followed by conducting saponification and imidation to PDIs may represent an alternative strategy to address the above concerns [56]." at the relevant place of the text (page#5, line#183 in the previous manuscript) and added a new reference#56 as suggested by the reviewer.

  1. The reviewer pointed out that the authors should consider extension of the title with the main focus of the PDIs extended/functionalized at the bay position.

              According to the comment, we revised the title of the manuscript by adding a new description "through Bay-Functionalization Strategy" at the end of the title given in the previous manuscript.

  1. The reviewer pointed out that the authors should give some information on the other examples found for liquid crystalline PDIs to produce noticeable impact of the core deformation on solubility.

              According to the comment, we revised the title of the manuscript by adding a new description "as has also been observed for liquid crystalline/mesogenic perylene dyes [68-70]" at the end of the related sentence (page#7, line#272 in the previous manuscript).

  1. The reviewer pointed out that the authors should summarize the FMO energy levels of materials in a separate figure.

              According to the comment, we inserted a new sentence "According to the literature data, schematic energy level diagrams of the donor and PDI materials used in the OPV devices are represented in Figure 6." at the relevant place of the text (page#21, line#781 in the previous manuscript) and added a new Figure 6 (page#22, line#808 in the previous manuscript).

  1. The reviewer pointed out that the authors should correct the description with regard to the lowest Vloss record of 3.4E.

              According to the comment, we revised the relevant description "representing the lowest level ever found for OSCs." (page#19, line#720 in the previous manuscript) with "representing the lowest level for OSCs at the time [127,128]" and added new references#127, 128 as suggested by the reviewer.

  1. The reviewer pointed out that the authors should correct typos and grammatical errors found in the manuscript.

              According to the comment, we corrected all the typos and grammatical errors as suggested in the attached manuscript by reviewers.
